Phylogenetic diversity of plants alters the effect of species richness on invertebrate herbivory

Dinnage Russell russell.dinnage@utoronto.ca
Ecology and Evolutionary Biology, University of Toronto , Toronto, ON , Canada
Cowling Richard
Electronic publication date: 2013 Jun 25
Publication date: 2013
Volume: 1
Electronic Location ID: e93
Received 2012 Dec 5; Accepted 2013 Jun 4
Copyright: © 2013 Dinnage
Copyright year: 2013
Copyright holder: Dinnage
License: This is an open access article distributed under the terms of the Creative Commons Attribution License, which permits unrestricted use, distribution, and reproduction in any medium, provided the original author and source are credited.
License URL: https://creativecommons.org/licenses/by/3.0/

Keywords: Cladophage, Ecosystem function, Biodiversity, Community ecology, Old fields, Herbivory, Phylogenetic diversity, Plant-Insect interactions

Funding: NSERC CGS-D grant I was generously supported by an NSERC CGS-D grant. The funders had no role in study design, data collection and analysis, decision to publish, or preparation of the manuscript.

==============================
Long-standing ecological theory proposes that diverse communities of plants should experience a decrease in herbivory. Yet previous empirical examinations of this hypothesis have revealed that plant species richness increases herbivory in just as many systems as it decreases it. In this study, I ask whether more insight into the role of plant diversity in promoting or suppressing herbivory can be gained by incorporating information about the evolutionary history of species in a community. In an old field system in southern Ontario, I surveyed communities of plants and measured levels of leaf damage on 27 species in 38 plots. I calculated a measure of phylogenetic diversity (PSE) that encapsulates information about the amount of evolutionary history represented in each of the plots and looked for a relationship between levels of herbivory and both species richness and phylogenetic diversity using a generalized linear mixed model (GLMM) that could account for variation in herbivory levels between species. I found that species richness was positively associated with herbivore damage at the plot-level, in keeping with the results from several other recent studies on this question. On the other hand, phylogenetic diversity was associated with decreased herbivory. Importantly, there was also an interaction between species richness and phylogenetic diversity, such that plots with the highest levels of herbivory were plots which had many species but only if those species tended to be closely related to one another. I propose that these results are the consequence of interactions with herbivores whose diets are phylogenetically specialized (for which I introduce the term cladophage), and how phylogenetic diversity may alter their realized host ranges. These results suggest that incorporating a phylogenetic perspective can add valuable additional insight into the role of plant diversity in explaining or predicting levels of herbivory at a whole-community scale.

Introduction

The insect herbivore flies, crawls, hovers, and feeds in a matrix of stems, roots, flowers and leaves. Thousands do this in a typical community of plants in what – to us – is an impenetrably complex tangle of traits, each the product of millions of years of evolution. How then are we to understand the aggregate patterns that emerge from such a community? Indeed, community ecologists have long struggled to understand how the identity and traits of individual plant species add together and interact to determine herbivore communities and the damage they cause.

Though much work on insect herbivory has focused on interactions between single pairs of insect and plant species, it is also important to understand the phenomenon at the level of whole plant communities. After all, many of the consequences of major ecosystem changes will manifest at the level of the community, and one-to-one interactions may not always be additive (Agrawal, Lau & Hamback, 2006; Agrawal et al., 2007).

One way of understanding herbivory at the community level is to look at the effects of plant species richness. Plant species richness is one of the simplest way to summarize information about many species at the community level, and studies of its role in herbivory link up nicely with the body of literature on the relationship between diversity and ecosystem functioning (Siemann et al., 1998; Haddad et al., 2001; Loreau et al., 2001; Hooper et al., 2005; Cardinale et al., 2006; Balvanera et al., 2006; Cardinale et al., 2007; Scherber et al., 2010a). Insect herbivore damage is a useful reflection of the activities of insect herbivores in plant communities, as well as a phenomenon of interest in and of itself, with implications for the health of a plant community.

There are some clear theoretical predictions about the effects of plant species richness on herbivory, which derive from a simple understanding of direct trophic interactions between plants and insects. Root (1973) suggested that increased plant species richness should lead to a decrease in specialist herbivores, and thus to a decrease in herbivory. The Resource Concentration Hypothesis states that if the number of plant species in a patch is higher, the density of each plant species must be lower, and so specialist herbivores of those species will be less likely to find such a patch, stay in such a patch, and do damage in such a patch. On the other hand, the presence of a wide variety of plant species provides a wide variety of nutritional resources for generalist herbivores, which may prefer to eat in diverse patches, and may thrive more in them too (Unsicker et al., 2008; Schuldt & Baruffol, 2010). I will refer to this idea as the Dietary Mixing Hypothesis (after Bernays et al., 1994).

Unfortunately, previous empirical literature has not provided a clear picture of which of these two hypotheses applies more widely to plant-herbivore systems, because whether plant species richness has a positive, negative, or no relationship with herbivory depends on the system under study. A review of the literature shows that there are roughly equal numbers of studies showing that plant diversity has a positive effect on herbivory as there are showing a negative effect, at least for studies that looked at the effect of species richness on the magnitude of herbivory (Table 1). This suggests that the role of plant species richness may sensitively depend on aspects of either the plant or the herbivore community. Based on the Resource Concentration Hypothesis and the Dietary Mixing Hypothesis described above, the relative abundance of generalist vs. specialist herbivores likely plays a role. For example, Lau et al. (2008) were able to separate damage done by specialists and damage done by generalists on Lespedeza capita planted in low and high diversity plots. They found that damage by generalists increased with high plant diversity and that damage from specialists decreased with high plant diversity, in accordance with the predictions of the Dietary Mixing Hypothesis and the Resource Concentration Hypothesis, respectively.

Table 1 Literature review of previous studies on the role of plant diversity in promoting or supressing herbivory.

Type refers to whether the species richness was manipulated (Experimental), natural species richness variation was taken advantage of (Observational), or the conclusion was based on estimates from previous literature (Meta-analysis). No study from the meta-analyses were repeated in this table. Consumer effect refers to whether the study estimated a measure of the amount of interaction between plants and consumers (Magnitude) – usually a damage measurement, or estimated the effect that interaction had on the plants’ fitness (Impact) – usually by measuring biomass in the presence or absence of herbivory. A positive relationship means that the study found that the magnitude or impact of herbivory increased with increasing plant diversity, a negative relationship means the study found that herbivory decreased with plant diversity. None means the study found no relationship between herbivory and plant diversity.

Study	Type	Species richness range	Measured	Measured on effect	Consumer	Relationship	
Lau et al. (2008)	Experimental	1 vs. 16	Leaf damage from generalists	1 Legume sp	Magnitude	Positive	
Mulder & Huss-Danell (2001)	Experimental	1, 2, 4, 8, 12	Leaf damage	Community	Magnitude	Positive	
Plath et al. (2011)	Experimental	1 vs. 3	Leaf damage	1 Rosea tree	Magnitude	Positive	
Prieur-Richard, Lavorel & Linhart (2002)	Experimental	3, 6, 18	Leaf damage	2 Conyza spp.	Magnitude	Positive	
Scherber et al. (2006)	Experimental	1, 2, 4, 8, 16, & 60	Leaf damage	All species + 3 phytometer spp	Magnitude	Positive	
Schuldt & Baruffol (2010)	Observational	25–68	Leaf damage	10 tree spp	Magnitude	Positive	
Vehviläinen, Koricheva & Ruohomäki (2007)	Meta-analysis	1 vs. Several	Leaf damage	2 tree spp (Oak & Alder)	Magnitude	Positive	
Wang et al. (2010)	Observational	1, 2, 4, 6, 8, 11	#plants grazed by sheep	Community	Magnitude	Positive	
Hanley (2004)	Experimental	3, 6, 12	Leaf damage by Mollusks	All spp	Magnitude	No	
Sobek et al. (2009)	Observational	Natural range across German deciduous forest	Leaf damage by insects	2 Maple spp	Magnitude	No	
Yguel et al. (2011)	Observational	Natural range across managed French forest	Leaf damage	1 Oak sp.	Magnitude	No	
Jactel & Brockerhoff (2007)	Meta-analysis	1 vs. Several	Leaf damage	Many individual spp.	Magnitude	Negative	
Lau et al. (2008)	Experimental	1 vs. 16	Leaf damage from specialists	1 Legume sp	Magnitude	Negative	
Massey et al. (2006)	Experimental	1 vs. 5	Leaf damage	1 Shorea sp	Magnitude	Negative	
McNaughton (1985)	Observational	Natural range across Serengeti	% Biomass consumed	Community	Magnitude	Negative	
Petermann et al. (2010)	Experimental	1, 2, 3, 4, 6, 9	Infestation by aphids	Community	Magnitude	Negative	
Sobek et al. (2009)	Observational	Natural range across German deciduous forest	Leaf damage by insects	1 Beech sp	Magnitude	Negative	
Unsicker et al. (2006)	Observational	18–45	Leaf damage	Many individual spp.	Magnitude	Negative	
Vehviläinen, Koricheva & Ruohomäki (2007)	Meta-analysis	1 vs. Several	Leaf damage by insects	1 Birch sp	Magnitude	Negative	
Wilsey & Polley (2002)	Experimental	1–4 (manipulated Evenness)	Infestation by spittlebugs	Solidago canadensis	Magnitude	Negative	
Mulder & Huss-Danell (2001)	Experimental	1, 2, 4, 8, 12	Reduction of biomass in insecticide plots	Community	Impact	Positive	
Scherber et al. (2010b)	Experimental	1, 2, 4, 8, 16, 60	% reduction of biomass in insect exclusions	Community	Impact	No	
Hillebrand & Cardinale (2004)	Meta-analysis	1–60	Absolute and % reduction of bio-mass in grazer exclusion	Community of phytoplankton	Impact	Negative	
Lanta (2007)	Experimental	1, 2, 4, 6	Reduction of biomass in Mollusk exclusions	Community	Impact	Negative	
Narwani & Mazumder (2010)	Experimental	1 vs. 4	Biomass consumed	Community of phytoplankton	Impact	Negative	
Pfisterer, Diemer & Schmid (2003)	Experimental	1, 2, 4, 8, 32	% reduction of biomass in insect exclusion	Community	Impact	Negative	
Stein et al. (2010)	Observational	15–37	% reduction of biomass in insect exclusions	Community	Impact	Negative	

Species richness is only a coarse measure of the diversity and structure of a community, and so including other sources of information about communities may help to clarify such simple predictions. Attempts to include more of the natural complexity of plant communities to understand community herbivory include using functional diversity in addition to plant species richness (Siemann et al., 1998; Koricheva et al., 2000; Prieur-Richard, Lavorel & Linhart, 2002; Scherber et al., 2006; Scherber et al., 2010b) reducing species composition to low dimensional quantitative measures using ordination (Koricheva et al., 2000; Pfisterer, Diemer & Schmid, 2003; Schaffers et al., 2008), and incorporating information about plant species’ evolutionary history, using community phylogenetics. Here, I explore the use of community phylogenetics to understand insect herbivory at the plant community level.

The increasing availability of phylogenetic information on plants and the recent development of sophisticated ways of incorporating this information into diversity measures (Helmus et al., 2007; Cavender-Bares et al., 2009; Vamosi et al., 2009; Cadotte et al., 2010; Pausas & Verdú, 2010), is opening a new avenue of exploration for understanding the effects of plants on ecosystem function at the whole community scale. For example, recent analyses have shown that various measures of phylogenetic diversity are associated with the productivity of plant communities (Cadotte, Cardinale & Oakley, 2008; Cadotte et al., 2009; Connolly et al., 2011; Flynn et al., 2011) and the insect assemblages associated with them (Dinnage et al., 2012). Here, I show that incorporating phylogenetic diversity into hypotheses about herbivory can provide a useful framework for predicting when species richness may increase or decrease herbivory in plant communities.

Phylogenetic diversity may be particularly important to herbivores and herbivory because of the widespread occurrence of herbivores with phylogenetically restricted diets – they feed on a group of closely related species (Ehrlich & Raven, 1964; Janzen, 1980; Ødegaard, Diserud & Østbye, 2005; Weiblen et al., 2006; Gossner et al., 2009; Futuyma & Agrawal, 2009). I will refer to these species as cladophages, from the greek klados for ‘branch’, and the greek phagein for ‘to eat’ – thus ‘branch-eater’, or an organism that feeds preferentially on just one of the branches of the tree of life that is available to them. The act of feeding in this manner can be called cladophagy. Ultimately, because many plant traits are conserved through evolutionary time, phylogeny will often be a reasonable proxy for phenotypic divergence – and thus phylogenetic diversity a reasonable proxy for phenotypic diversity (Cavender-Bares et al., 2009). Since herbivores are presumed to feed on phenotypically similar plant species more often than not, phylogenetic diversity will indirectly effect how herbivores interact with plant communities.

In this study I asked whether the phylogenetic diversity of old-field plant communities can predict the amount of herbivore damage on plants at a community-scale.

Materials and Methods

Data collection

The study was conducted at the Koffler Scientific Reserve (KSR) at Joker’s Hill (King City, Ontario, Canada; http://www.ksr.utoronto.ca/jh.html), a 350-hectare property containing a mix of primary forest, secondary forest and open habitats, including a large area of old field sites. I was granted permission by the site director – Ann Zimmerman – to conduct the research there.

Species survey

I haphazardly selected 38 10 × 10 meter plots spread across the available old field habitat at KSR. In early August, I surveyed each plot within 4 randomly placed 1 × 1 meter quadrats, one in each of the four quadrants of the larger plot. I noted the presence or absence of each forb species within the quadrats and combined the data from all four quadrats into a low resolution measure of abundance (ranging from 0–4) for each species within each 10 × 10 meter plot. I decided to focus on forbs to the exclusion of grasses for several reasons. Herbivory from chewing insect is difficult to measure on grasses but is obvious on forbs. There were only two common species of grass, which occurred in nearly every one of the plots surveyed, and so inclusion of grasses in the phylogenetic diversity measures would have only diluted the signal from the forbs. I hypothesized that phylogenetic diversity amongst forbs would be the most important to understanding herbivory on forbs. This is because grasses and forbs share few herbivores, except for the most generalist. As an example, Dinnage et al. (2012) showed that the largest difference in arthropod communities collected from different prairie plant species was between grasses and forbs.

Using this data, I calculated the species richness and abundance-weighted phylogenetic diversity for all 38 plots. This data, along with the species composition of all 38 plots, can be found in Table S1 in Supplemental Information.

Herbivory survey

In order to quantify the amount of herbivore damage in the plots, I selected 27 of the most common species at KSR to measure damage (Fig. 1). These species acted as phytometers to capture the overall herbivore pressure in the plot. In late August – over a period of approximately two weeks – I measured herbivore damage rates in the 38 plots. For each of the phytometer species that occured in a given plot, I sampled 10 individual plants by counting the number of damaged leaves and the number of total leaves on each one. The dataset then consisted of 1862 datapoints, each from an individual plant.

Figure 1 Phylogenetic tree with branch lengths representing all species in this study.

Species highlighted in grey are species for which herbivore damage measurements were taken. Branch lengths represent divergence times. Methods for generating the tree can be found in Dinnage (2009).

I chose this set of phytometer species before the commencement of the study, based on my personal knowledge of which species seemed to make up the most apparent membership of the old field communities (i.e., the ‘overstory’). This intuition seems justified, because of the 25 species which were surveyed in the plots, but for which I did not measure damage, 17 were found in only 1 or 2 plots (<5% of total plots), and 24 were found in 5 or fewer plots (<13% of total plots). It is unlikely that measurements of herbivory on these species could have changed the results presented here substantially, unless rare species systematically differed in their response to plant diversity. The one exception to this general pattern was Taraxacum officianale, which was found in 16 plots and was not measured for herbivory. However, T. officianale is an understory plant, which remains as a rosette for most of its life history. It only produces a few leaves (4–10), all of which usually sustain some herbivore damage (personal observation). Given this lack of variation in herbivory within T. offianale under the method of measurement that I chose, it would be unlikely to affect the results substantially (because if there is no variation, then it cannot vary with plant diversity or anything else). For full information on species abundances across the 38 plots, see Table S1.

Phylogeny and diversity indices

I created a phylogenetic tree containing all the most common old field species found at my site using methods previously described (Dinnage, 2009). I then pruned the tree so that the remaining species matched those found in the plots used for this study (Fig. 1).

Using this phylogeny I calculated an index of abundance-weighted phylogenetic diversity for each plot. There are several such indices available – I chose Phylogenetic Species Evenness (PSE; Helmus et al., 2007). PSE measures the amount of evolutionary history represented in a community by calculating the expected variance in a hypothetical continuous trait modeled as evolving through Brownian motion across the community phylogeny, and is standardized by the expected variance of the same modeled trait on a star phylogeny – where phylogenetic distances among all species are equal. This captures one of the fundamental features of interest in phylogenetic diversity: that it may encapsulate information about the ecological similarity of species in a community. I chose PSE for this reason, and because it is theoretically (and empirically in this study) independent of species richness. Thus, by using it, I could evaluate the independent contributions of species richness and phylogenetic diversity without the problems of co-linearity and heteroscedasticity common to many other phylogenetic diversity indices (e.g., Schweiger et al., 2008; Pio et al., 2011). PSE ranges between 0 and 1, where high values are associated with high phylogenetic diversity. Low values are associated with low phylogenetic diversity, or phylogenetic clustering – small average phylogenetic distances among species in the community. It is possible that I could have seen different results with a different metric, but PSE is highly correlated with a number of other metrics (Cadotte et al., 2010), and so the results presented here should at least apply to this general ‘class’ of phylogenetic diversity indices.

I used species richness rather than an abundance-weighted measure because I was interested in the effects of adding or subtracting species. This combined with the phylogenetic diversity measure can give a statistical measure of adding species of different phylogenetic distances from the species already present. In addition, abundance weighting is easily interpretable for a measure like phylogenetic diversity, because in essence it is a weighted average, and phylogeny is made from continuous measures of branch length. On the other hand, abundance weighted measures like the Shannon-Weaver index are abstracted quantities with no simple interpretation biologically (e.g., Goodman, 1975; Austin, 1999). I did calculate evenness for the plots and it was not correlated with phylogenetic diversity, and so the phylogenetic diversity measure, though abundance-weighted, was not confounded with species evenness. In addition, evenness was not a significant predictor of herbivory when included as a factor in the model described below, and so I did not use it.

Statistical analysis

In order to account for variation in species composition across plots, I used a generalized linear mixed model (GLMM) with crossed random effects, to disentangle species and plot-level effects on herbivory. The number of damaged leaves on a plant can be modelled as a binomial distribution, with a damaged leaf considered a Bernoulli success (for the herbivores) and an undamaged leaf a failure. Factors that may influence the probability of a leaf being damaged were incorporated into the model with a logit link.

I used a logit-normal-binomial statistical model to analyze the data, as follows: Damagedi∼Binomial(pi,Ni)

Logit(pi)=α+μspecies[i]1+μplot[i]2+μobs[i]3+βpdPDplot[i]+βsrSRplot[i]+βpd∗sr(PD∗SR)plot[i]+βdateDATEplot[i]

μspecies[i]1∼Normal(0,σspecies2),μplot[i]2∼Normal(0,σplot2),μobs[i]3∼Normal(0,σobs2)

where Damagedi is the number of damaged leaves on individual plant i, pi is the probability of any given leaf on plant i being damaged, and Ni is the total number of leaves on plant i. PDplot[i] and S Rplot[i] are the phylogenetic diversity and the species richness, respectively, for the plot in which plant i was found. DATEplot[i] is the date on which I sampled the plot, to control for any increases in herbivory that may have occurred while the sampling was ongoing. μspecies[i]1 and μplot[i]2 are the random effects for species and plot, respectively. μobs[i]3 is a random effect for the individual observation which was included to account for any over- or under-dispersion in the data since the binomial distribution contains no variance parameter. The α (intercept) and β parameters are the fixed effects.

I used the lme4 package (Bates, Maechler & Bolker, 2011) for R statistical environment (R Development Core Team, 2010) to fit the model.

Plot level herbivory (after accounting for species-level variation, sampling date, and observation) can then be estimated as the best linear unbiased predictors (BLUPs) – sometimes referred to as the conditional modes – of pi (hereafter estimated proportional leaf damage). For plot j, this is equal to pj=Logit−1(α+μj2+βpdPDj+βsrSRj+βpd∗sr(PD∗SR)j). Likewise, herbivory of species k (after accounting for plot-level variation, sampling date and observation) is pk=Logit−1(α+μk3). These values were used for plotting (Figs. 2 and 3).

Figure 2 Species-level herbivory of the 27 species for which herbivore damage measurements were taken.

On the left is the phylogenetic relationships of the species with branch length representing time since divergence. On the right is a bar chart whose bars represent the best linear unbiased predictors (BLUPs) or conditional modes of the estimated proportional leaf damage for each species. Error bars are based on the conditional variance-covariance matrix generated by the model fitting procedure (lmer function in the lme4 package for R) and are conditional mode + /− 1 conditional standard deviation.

Figure 3 Two figures showing the relationship between estimated proportional leaf damage and plot-level plant species richness and phylogenetic diversity.

(A) Points represent best linear unbiased predictors (BLUPs) or conditional modes of plots. The size of the points is proportional to the species richness of the plot so that large points are speciose communities. Error bars are based on the conditional variance-covariance matrix generated by the model fitting procedure (lmer function in the lme4 package for R) and are conditional mode + /− 1 conditional standard deviation. Fitted lines are back-transformed predicted values from the full generalized linear mixed model, for four different pre-set values of species richness (3, 6, 12, & 17) representing the full range of species richnesses in this study. (B) A heatmap which shows the back-transformed fitted surface of the full generalized linear mixed model. Points are the plot BLUPs, their colour represents their value as per the legend. Points that are darker than the surrounding colour fall below the predicted surface; points which are lighter fall above it.

All continuous predictor variables were centred by subtracting their means prior to analysis, so that the main effect of the variable refers to its slope at the mean of all other variables.

Significance of the relationships between estimated proportional leaf damage and the fixed factors was determined using a parametric bootstrap approach. For each fixed factor, data was simulated under a simplified model without the fixed factor of interest. The full model was then fit to the simulated data and the z statistic was calculated. The observed z statistic from the full model was then compared to the distribution of z values obtained from 1000 such simulations, and a p value determined as the percent of simulated z values whose absolute value was greater than or equal to the absolute value of the observed z-value (Pinheiro & Bates, 2000).

Results

The average percentage of leaves damaged in this study was 53.6%. The species of the plant had a large effect on the estimated proportional leaf damage, with estimated variance for the species random effect of 2.4 compared with 0.26 for the plot random effect and 0.82 for the observation random effect (equivalent in this model to residual variance). Leaf damage rates ranged from 5% for the non-native highly defended Euphorbia cyparissias to close to 98% for the (also non-native) forage legume Medicago sativa (Fig. 2). Most of the common, native species such as Solidago canadensis, Asclepias syriaca and Symphyotrichum spp. had intermediate to high levels of herbivory (Fig. 2).

Sampling date was positively related to estimated proportional leaf damage but not significantly so (z = 1.05, p = 0.334; Table 2).

Table 2 Statistics for the fixed effects.

Statistics for the fixed effects in a generalized linear mixed model (GLMM) with binomial family and logit link. P r(|z|obs < |z|sim) is the p-value generated from a parametric bootstrap on the z values.

Parameter	Name of factor	Estimate	z value	Pr(|z|obs < |z|sim)	
α	Intercept	0.15			
βdate	Sampling date	0.10	1.05	0.334	
βsr	Plant species richness	0.23	2.26	0.042*	
βpd	Plant phylogenetic diversity (PSE)	−0.19	−2.10	0.055	
βsr∗pd	Plant diversity interaction	−0.28	−2.27	0.043*	
Notes.

* z value is significant at the alpha = 0.05 level.

After accounting for species composition, I found a significant positive main effect of species richness (z = 2.26, p = 0.042), and a near-significant negative main effect of phylogenetic diversity (z = −0.19, p = 0.055) on herbivory at the plot-level (Table 2, Fig. 3). There was also a significant negative interaction between phylogenetic diversity and species richness of plants (z = −2.27, p = 0.043; Table 2, Fig. 3), so that the positive effect of species richness on herbivory decreased with increasing phylogenetic diversity. This means that the plots with the highest estimated proportional leaf damage were plots with many species which tended to be closely related. Plots with few species and plots with more distantly related species tended to have lower herbivory (Fig. 3). Plots with the lowest phylogenetic diversity and the highest species richness had about twice as much proportional leaf damage (∼80%) as plots with high phylogenetic diversity and low species richness (∼40%, Fig. 3).

Discussion

The results of this study show that phylogenetic diversity is a useful measure for predicting the level of herbivory in plant communities. I have shown that community-level herbivory tends to increase with plant species richness, consistent with some past studies, and in contrast to other past studies (Table 1). On the other hand I also showed, for the first time, that high phylogenetic diversity of plant communities is associated with overall lowered levels of herbivory. More importantly, there was an interaction between plant species richness and phylogenetic diversity, such that with increasing phylogenetic diversity, the positive effects of species richness on herbivory decrease. This means that the effects of plant species richness on herbivory is dependent on the level of phylogenetic diversity in the plant community (and vice versa).

The effects of plant species richness on herbivory

The positive effect of plant species richness on herbivory (at the mean level of phylogenetic diversity) found here is the opposite of early ecological predictions (Elton, 1958; Root, 1973), but is not surprising given the frequency with which this pattern has been found in other systems (e.g., Mulder & Huss-Danell, 2001; Prieur-Richard, Lavorel & Linhart, 2002; Scherber et al., 2006; Vehviläinen, Koricheva & Ruohomäki, 2007; Lau et al., 2008; Schuldt & Baruffol, 2010; Plath et al., 2011; see Table 1). Theory predicting decreased levels of herbivory in speciose communities was based on the idea that specialist herbivores would be less likely to find and more likely to abandon diverse patches because abundance of their preferred hosts would necessarily be low in such communities (the Resource Concentration Hypothesis: Root, 1973). However, this theory is dependent on the herbivore species’ being monophagous. Thus, in systems where monophages are rare or do not constitute the most damaging class of herbivores, the Resource Concentration Hypothesis is unlikely to apply. Another theory for why herbivore abundance and thus herbivory may be low in speciose plant communities is the Enemies Hypothesis (Elton, 1958; Root, 1973), which suggests that predators will be more abundant in diverse plant communities, and they will suppress herbivore populations. However, several recent empirical tests of this hypothesis have shown little direct relationship between plant species richness and predator abundance (Scherber et al., 2010a; Schuldt et al., 2011).

Theory for potential causes of a positive relationship between species richness and herbivory, on the other hand, is less developed and less frequently cited. Some authors have suggested that this may be the result of nutritional advantages to generalist herbivores who have access to a variety of food resources in diverse patches (Bernays et al., 1994; Unsicker et al., 2008; Schuldt & Baruffol, 2010), which I refer to here as the Dietary Mixing Hypothesis. Other authors have suggested that spillover from preferred to less preferred host species could explain the effect, sometimes referred to as ‘associational susceptibility’ (White & Whitham, 2000).

On the other hand, the relationship between plant species richness and herbivory depends on the level of plant phylogenetic diversity you measure it at (the main effect is measured at the mean level of phylogenetic diversity – a natural place to do so). The implication of the significant interaction I found between plant species richness and plant phylogenetic diversity in explaining herbivore damage is that the phylogenetic diversity of the plots alters how species richness affects herbivory. I devote the next section to explaining how this interaction can be potentially explained as a natural outcome of a few simple principles.

The effects of plant phylogenetic diversity on herbivory (as mediated through plant species richness)

The likelihood of the Resource Concentration Hypothesis, the Dietary Mixing Hypothesis, or associational susceptibility being a factor may be related to the phylogenetic structure of the plant community, because many herbivore species, and especially insect herbivores, have phylogenetic structure in their diet – they feed on few or many species which tend to be closely related (Ehrlich & Raven, 1964; Janzen, 1980; Ødegaard, Diserud & Østbye, 2005; Weiblen et al., 2006; Gossner et al., 2009; Futuyma & Agrawal, 2009). And so theory which is based on the dichotomy of specialist vs generalist herbivores is likely to be too simplistic to adequately describe real systems. Instead it is more useful to use three categories of herbivore which may inhabit a plant community: (1) True specialists or monophages – which feed on only a single plant species, (2) Phylogenetic specialists or cladophages – which feed on a group of related plant species, and (3) true generalists or polyphages – which feed on a group of plant species which has no pattern with respect to phylogeny at some relevant phylogenetic scale.

Though some authors use the term specialist and oligophage in a way consistent with the use of cladophage here – i.e., it is implied or explicitly stated that the author is using those terms to refer to species that feed on more than one closely related species – these terms are also used in other ways. Oligophage literally means a species that feeds on a ‘few’ hosts (oligo- is latin for ‘few’), but does not specify whether these species are poly-, para-, or mono-phyletic. And the term ‘specialist’ usually needs qualification – e.g., a frugivore can be called a fruit specialist, and a cladophage can be called a phylogenetic specialist. It may then be tempting to conclude that I am introducing the term ‘cladophage’ because I am tired of typing out the much longer ‘phylogenetic specialist’ repeatedly. There is value to concision, but I hope the term cladophage will be useful for more than this.

The term cladophage can be more precisely defined as a species whose diet consists of other organisms that are more closely related than expected by chance. This means there is some vagueness to the term, as there are multiple ways to decide what is expected by chance (two methods for doing so can be found in Ødegaard, Diserud & Østbye, 2005; Weiblen et al., 2006), which depend to some extent on what one is considering the potential diet pool. For example, the phylogenetic scale of the diet pool is important because almost any species can be considered a cladophage at some phylogenetic scale. For example, all insect herbivores are cladophages at the scale of the tree of life, because they only eat plants – a monophyletic clade. Another example is Milkweed Beetles (Tetraopes spp.), which only feed on Milkweed plants (Asclepias spp.), and so are cladophages when considering all possible plants. On the other hand, if you are only interested in the Milkweed species as a potential diet pool, then Milkweed Beetles may not be considered cladophages because their diet could be random with respect to phylogeny within the Milkweed genera (Farrell & Mitter, 1998). And so to properly define these categories, the phylogenetic scale of interest must also be defined. Practically speaking, for the study of herbivory, this will usually be based on the phylogenetic tree containing all the plant species present in the study, which will usually be the most appropriate for understanding local ecological dynamics (Cavender-Bares et al., 2009). This definition then potentially includes species whose diets are paraphyletic, as well as monophyletic. It should be noted that other terms contain some such relativity, including monophage. That is, a monophage is defined as a species that only eats one other organism, but it can be measured as what a species is observed to eat in its natural habitat, or as what a species potentially could eat, if given the opportunity. A monophage in a particular habitat may not be a monophage in another where some new host species become available. This concept of ‘effective’ specialization is important to understanding why phylogenetic diversity might affect the relationship between host species richness and herbivory.

Using this framework we can begin to make hypotheses about the effects that phylogenetic diversity might have on herbivory in plant communities. It is clear that neither monophages nor polyphages should be affected by phylogenetic diversity of plants, at least directly – though nutritional effects are possible in generalists. On the one hand, cladophages are likely to respond to phylogenetic diversity of plants due to direct interactions with their host-plants (Dinnage et al., 2012). In a plant community with low phylogenetic diversity, if one host capable of supporting a particular cladophage is present, there is likely to be other suitable hosts as well (assuming the presence of at least moderate species richness). On the other hand, in plant communities with high phylogenetic diversity, if any hosts are suitable for a particular cladophage, it is likely to be the only one. This means that cladophages present in low phylogenetic diversity plant communities will be ‘effective polyphages’; cladophages present in high phylogenetic diversity plant communities will be ‘effective monophages’. Therefore, in plant communities with low phylogenetic diversity, cladophages are more likely to be able to take advantage of dietary mixing effects (as per the Dietary Mixing Hypothesis), and so species richness should have a positive effect. This is consistent with how phylogenetic diversity changed the effect of plant species richness on herbivory observed in this study – that is, the positive effect of species richness increased at low phylogenetic diversity, and decreased at high phylogenetic diversity (Fig. 3).

In plant communities with high phylogenetic diversity, cladophages will usually only be able to feed on one or a very few species, and so they may respond according to the Resource Concentration Hypothesis, that is, increasing the number of species in the plant community will make it more difficult for them to find their preferred host. Though in this study, the effect of species richness never became negative even at the highest level of phylogenetic diversity, it did become very nearly flat (Fig. 3). It is possible that the relationship would have become negative if there had been plots in this study with even higher phylogenetic diversity than observed.

There are several reasons why the relationship between plant species richness and herbivory may not become completely negative at the highest phylogenetic diversity of plants. For one, the effects of cladophages will be overlaid on the effects from monophages and polyphages. If the polyphage effect is more important in the system, then positive effects of plant species richness will likely predominate. Another possibility is that low species richness can sometimes have a negative effect on monophages, instead of the expected positive effect according to the Resource Concentration Hypothesis. These effects have been termed ‘resource dilution effects’ (Otway, Hector & Lawton, 2005), and the commonness of their occurrence has yet to be established.

Even though the effect of phylogenetic diversity on herbivory depended on plant species richness, the overall main effect – measured at the mean level of plant species richness – was negative, suggesting that in this system, there is an average decline in herbivory with phylogenetic diversity. Though no previous study has looked at the role of phylogenetic diversity per se on rates of herbivory in plant communities, my results are consistent with the results of Jactel & Brockerhoff (2007), who found in a meta-analysis that reductions in herbivory for focal trees grown in mixed stands instead of monoculture stands were stronger when the associated tree species in the mixed stand was taxonomically unrelated to the focal species.

The role of cladophages in ecological systems

The relative abundance of cladophages in this system is unknown, but it is likely that they occur regularly in most systems. It has been noted that the tendency for herbivores to feed on several closely related species is widespread (Ehrlich & Raven, 1964; Janzen, 1980; Futuyma & Agrawal, 2009). If so, it is predicted that insect herbivore assemblages should be more similar in closely related plant hosts than in more distantly related ones. This pattern has been found in several recent studies in different systems (Ødegaard, Diserud & Østbye, 2005; Weiblen et al., 2006; Gossner et al., 2009). For example, in a tropical system, Weiblen et al. (2006) found that approximately half of the herbivore species they studied could be categorized as phylogenetic specialists.

The widespread existence of cladophages is also an assumption underlying the hypothesis that invasive species which are more distantly related to native species should experience higher levels of enemy release and thus lower levels of herbivory (Mitchell et al., 2006). This pattern, too, has been found in several recent studies (Dawson, Burslem & Hulme, 2009; Hill & Kotanen, 2009; Ness, Rollinson & Whitney, 2011). Native species have also been shown to experience lower herbivory when growing with neighbours which are distantly related (Yguel et al., 2011; Ness, Rollinson & Whitney, 2011), further suggesting not only that cladophages are common, but that they exert important influence in ecological systems.

Alternative explanations

There are several alternative explanations for why phylogenetic diversity might indirectly affect herbivory. It has been suggested that phylogenetic diversity is a good proxy for unmeasured trait or functional diversity of a community (e.g., Cadotte, Cardinale & Oakley, 2008; Cavender-Bares et al., 2009). It is possible that functional diversity may be responsible for reduced herbivory. It is not clear why functional diversity would decrease herbivory directly, but it is possible that increased functional diversity could promote predator recruitment and abundance, which in turn could suppress herbivores. Functional diversity could increase the architectural complexity of a plot, allowing more hunting niches, nesting sites, and places to hide and stalk for predators (Andow & Prokrym, 1990; Coll & Botrell, 1996; Beals, 2006; Woodcock et al., 2007). For example, Dinnage et al. (2012) found that in an experimental prairie system, predator abundance increased strongly with phylogenetic diversity. However, in the old field system at my study site, there was little variation in structural forms. Most species grew with tall stalks, and leaves parallel to the ground along their entire height. There were a few understory species (rosettes), a few vines, and a few tree or shrub saplings. Most plots had all of these, regardless of phylogenetic diversity (Table S1, personal observation). These growth forms occurred across the phylogenetic tree without any obvious patterns. However, I cannot rule out this explanation without further information on the abundance of predators. The effect may also have been the result of a combination of factors, that is, both increased predator abundance and reduced diet breadth are jointly responsible for the decrease in herbivory.

It is possible that a correlation between phylogenetic and functional diversity contributes to the patterns I observed in addition to the effects of cladophagy, but it does not offer a good explanation for the interaction between phylogenetic diversity and species richness, or the effect of species richness itself. This is because functional diversity is also expected to – and usually observed to – increase with species richness, and in fact this is often the explanation offered for why species richness affects various ecosystem functions (Tilman, 1997; Loreau et al., 2001; Hooper et al., 2005; Cardinale et al., 2006; Balvanera et al., 2006; Duffy et al., 2007; Cadotte et al., 2009; Fornara & Tilman, 2009; Cardinale, 2011; Connolly et al., 2011). Given this, we should expect to see a synergy between phylogenetic diversity and species richness, rather than a negative interaction.

Though the observational nature of this study gives it the advantage of realism, it also means I cannot completely rule out all confounding factors that may explain my results. If, for example, there were abiotic conditions which influenced both species richness and phylogenetic diversity, and also affected herbivory, and which varied sufficiently across my sites, this could spuriously generate the observed correlations (Procheş et al., 2009). Though all the plots in this study came from a small geographic area and from a single habitat type, and thus likely do not vary much in environmental conditions, it is possible that some soil characteristics may have varied between the plots. Though environmental correlates of species richness have attracted much interest in the past, especially at large spatial scales, little is known about what environmental factors might influence phylogenetic diversity. Since it is often supposed that phylogenetically closely related species will compete more strongly, it is possible that plots with conditions that suppress competition amongst plants are the only ones where many species which are closely related can coexist. If such conditions also promoted high herbivory, then this could lead to high herbivory in plant communities with high species richness and low phylogenetic diversity, as observed. To fully understand the role of phylogenetic diversity in ecosystem function will require conducting large-scale experiments where phylogenetic diversity is explicitly manipulated, but observational studies in natural settings such as this will remain invaluable.

Limitations of this study: measuring the impact of herbivory

This study shows that many plants may receive lower herbivore damage in phylogenetically diverse communities. This result is similar to those of Yguel et al. (2011) who found that oak trees that grew amongst distantly related trees experienced lower herbivory. Yguel et al. (2011) suggested that this may produce a selective benefit that could promote the evolution of a strategy to grow amongst distant relatives. This is an intriguing possibility, but studies which only measure the magnitude of damage on plants cannot draw this conclusion, unless they also measure the impact of damage on the plants. In order to understand how selection might affect a plant’s preference for its phylogenetic neighbourhood, we need to know how the fitness of the plants are affected.

Damage is generally thought of as a negative impact on plant fitness, but many plants are able to tolerate large amounts of damage without sustaining large reductions in fitness (Rosenthal & Kotanen, 1994; Strauss & Agrawal, 1999). To understand fitness impacts of herbivory generally requires measuring plant performance in the presence and absence of herbivory. In Table 1, I show that although many studies that looked at the magnitude of herbivory found that plant species richness increased herbivory, all but two studies that looked at the impact of herbivory found a negative association with plant species richness, such that speciose communities had a smaller difference in biomass between herbivore exclusion and control treatments. This suggests that plant diversity may provide benefits in terms of a plant’s ability to tolerate herbivory which more than makes up for any increases in the actual amount of herbivore damage.

I therefore cannot conclude that any reduction in herbivory due to high phylogenetic diversity necessarily results in an advantage to plants growing with their distant relatives. However, the only way this would not be the case is if plants growing in low phylogenetic diversity communities were more tolerant of damage than plants growing in high phylogenetic diversity communities, and this increase in tolerance fully compensated for the increase in herbivore damage. This seems unlikely, but cannot be ruled out without herbivore exclusion studies. It is possible, for example, that plants growing with distant relatives will tend to be in more marginal habitat, since abiotic niche requirements also may be similar for close relatives (Cavender-Bares et al., 2009). Plants in marginal habitat may be impacted by herbivory more.

Implications for other studies

As I show in Table 1, the results of previous studies are inconsistent about the relationship observed between herbivory and species richness. The interaction I observed between phylogenetic diversity and species richness in this study may offer an explanation. That is, the degree and even direction of the relationship between species richness and herbivory may depend on the difference in phylogenetic diversities of the communities that are compared. Though I observed such a relationship at a local scale, between individual plots, it may also apply on a larger scale. For example, it is possible that I may have observed an overall positive main effect of species richness in my study because there was generally a low amount of phylogenetic diversity in this system compared with other systems. Old fields tend to be dominated by species in the family Asteraceae, and the one I studied is no exception. Other systems where the question has been studied may draw from a larger pool of evolutionary history, where negative effects may perhaps be more likely.

Of course, the relative abundance and importance of monophages, polyphages, and cladophages present and active in the plant communities may also alter the observed effect of plant diversity. For example, simplified ecosystems such as agricultural systems – where we often see negative diversity-herbivory relationships (Andow, 1991) – may be dominated by monophages, because monocultures on very large scales may discourage polyphages (or even many cladophages). Whereas polyphages need not be less common compared with monophages in monocultures that are imbedded in a more diverse landscape (as is the case for most biodiversity experiments), as long as the scale of the plots does not exceed the dispersal capability of the polyphages. In natural ecosystems, some systems may be more dominated by extinction-recolonization dynamics of both plants and herbivores, whereas others may be dominated by relatively stable populations of low-dispersal plant and herbivores, which could also have major effects of Resource Concentration and Dietary Mixing. All else being equal, I suggest that the phylogenetic or phenotypic diversity is likely to play a strong role, in addition to the other factors.

Conclusion

Here I have shown that incorporating measures of phylogenetic diversity can improve our understanding of the role of plant diversity in promoting or suppressing herbivore damage. Phylogenetic diversity was negatively correlated with herbivory in an old field system, and determined the degree to which species richness influenced herbivory as well. If we consider a reduction in herbivore damage as a positive outcome at the community scale, as is often the case in agricultural ecosystems, then we may wish to prioritize the conservation and restoration of phylogenetic diversity, perhaps even at the expense of some species richness.

Supplemental Information

Table S1 Plant community herbivory, species richness, phylogenetic diversity, and composition for 38 old field plots

Click here for additional data file.

I am grateful to Anna K. Simonsen for logistical support. Ann P. Zimmerman provided assistance and granted permission for the use of KSR. Peter Abrams provided valuable feedback on an earlier draft of the manuscript.

Additional Information and Declarations

Competing Interests

Author Contributions

Field Study Permissions

I have no competing interests.

Russell Dinnage conceived and designed the experiments, performed the experiments, analyzed the data, contributed reagents/materials/analysis tools, wrote the paper.

The following information was supplied relating to ethical approvals (i.e., approving body and any reference numbers):

I was granted permission from the director of the Koffler Scientific Reserve, Ann Zimmerman, to conduct the research on the property.

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
