# Peer review of "Phylogenetic diversity of plants alters the effect of species richness on invertebrate herbivory"

_PeerJ, doi:10.7717/peerj.93_

## Round 0.1 · original submission · Minor Revisions

Two reviewers have read your manuscript and both find it acceptable with relatively minor revision. Congratulations! Both reviewers expressed a need for more information and/or clarification regarding some issues (see reviews) and both had concerns about the term cladophage.

Reviewer 1 ·

Basic reporting

The manuscript is beautifully written, and it addresses a very interesting topic. It is in fact a first test for the usefulness of a fairly novel method of measuring biodiversity (phylogenetic diversity) in assessing a fascinating community ecology puzzle.

I do however get the sense that the author is trying his best to keep it simple, possibly simpler than the story should be. Some necessary information is simply not provided. How were forb species selected for damage measurements? Why only use one measure of phylogenetic diversity? Would others yield similar results? If not, should we be using them at all in other studies? These matters should be explained.

Experimental design

Further information is needed to understand how species were selected for damage surveys among all the forbs present. The author says “some of the most common”. Maybe present some occupancy data – these can even be added on a combined figure (Fig1-2), as a filter layer between the phylogeny and the damage data. Then lower down discuss how this method of selection may or may not have affected your results. After all, excluding rare species could tip the balance between the ‘dietary mixing’ and ‘resource concentration’ hypotheses – which is the very crux of the paper.

Validity of the findings

Important: Line 238 “near-significant” rather than “marginally significant” - this is actually the result that the entire paper hangs on.

Additional comments

I agree that “cladophages” makes more sense than “oligophages”. Oligo- refers to species, and these could form mono- para- or poly-phyletic groups. Then again, my guess would be that the plants a herbivore eats are more often than not forming paraphyletic groups, with key species in the clade developing some form of resistance and becoming subsequently unsuitable. Maybe discuss a few more potential limits of this new term.

‘Implications for other studies’ section. The results are from a Canadian old field system, and the author does make some hints as to how they may compare across systems. But for a minute forget agriculture and other anthropogenic disturbance. Even if we stick to natural systems, some are dominated by plant and insect species with extinction-recolonisation dynamics; some by stay-at-home-ers (or at least by different degrees in between the two). This would surely influence the importance of ‘dietary mixing’ vs. ‘resource concentration’. Place this bit in a broader context.

Some additional references could provide support in a few places: lines 176-177 (there are a few to choose from!); 398-401 (maybe Proches et al 2009 MPE) ; 457-458 (Novotny et al 2006 Science).

Minor: Line 34: while I appreciate the literary effect of the opening lines, not all insect herbivores fly and hover. Specialists generally have to, but many generalists are flightless.

Reviewer 2 ·

Basic reporting

no comments

Experimental design

no comments

Validity of the findings

no comments

Additional comments

Somewhere in the introduction, you may want to be specific about why phylogeny and phylogenetic history could be an important predictor of damage.

You introduce the term cladophage, which is cute, but in my opinion, reduces emphasis on the main point, which I think is the following. Phylogenetic similarity is useful here because it is a proxy for phenotypic similarity… that is, we expect that herbivores eat highly similar things, and since we cannot measure the phenotype in its entirety, and yet the phenotype, on average, will be proportionally similar to phylogenetic distance among species, it is a reasonable proxy. I suggest adding one sentence to the introduction stating something to this effect (although you will need to be more clear than I was ;-). Along these lines, the term specialist has long been equated to cladophage in studies of herbivory (see book by Schoohoven et al.), another reason to not introduce the term. Yes, monophages are a separate category, but there are so few of those types of specialists (and we already have a name for them too).

The last sentence of the introduction should be expanded to include the full set of questions addressed here (presumably to include the literature review, and the test of richness vs. phylogenetic diversity – and their interaction).

Line 130, add a comma after grass (I think this is what the author means).

The paper overall is claims a “main effect” of both richness and phylogenetic diversity, and then adds “even more importantly” that there is an interaction. I wonder if this is not stated as clearly as possible. In other words, many statisticians would argue that the main effects are not interpretable if there is an interaction. Yes, in this case, the main effect of both factors are statistically significant, but the interactions directly indicates that the interpretation of each factor depends on the value of the other. I would encourage the author to take this more sophisticated view of the results (esp in the first paragraph of the discussion, but throughout the paper).

I would have liked to see a paragraph that specifically addresses potential mechanisms of the interactive effects observed. I was left a little hungry for that.

---

## Round 0.2 · accepted · Accept

I enjoyed reading the rebuttal, especially the bit on interpreting interactions among significant main effects.

I am happy with your thoughtful responses to the reviewers. The paper is now acceptable for publication in PeerJ.